# Group B Streptococcus: Virulence Factors and Pathogenic Mechanism

**DOI:** 10.3390/microorganisms10122483

**Published:** 2022-12-15

**Authors:** Yuxin Liu, Jinhui Liu

**Affiliations:** 1Queen Mary School, Nanchang University, Nanchang 330006, China; 2Biomedical Sciences School, Queen Mary University of London, London E1 4PD, UK; 3School of Basic Medical Sciences, Nanchang University, Nanchang 330006, China

**Keywords:** Group B Streptococcus, virulence factors, vaginal colonization, perinatal infection, vaccine

## Abstract

Group B Streptococcus (GBS) or *Streptococcus agalactiae* is a major cause of neonatal mortality. When colonizing the lower genital tract of pregnant women, GBS may cause premature birth and stillbirth. If transmitted to the newborn, it may result in life-threatening illnesses, including sepsis, meningitis, and pneumonia. Moreover, through continuous evolution, GBS can use its original structure and unique factors to greatly improve its survival rate in the human body. This review discusses the key virulence factors that facilitate GBS invasion and colonization and their action mechanisms. A comprehensive understanding of the role of virulence factors in GBS infection is crucial to develop better treatment options and screen potential candidate molecules for the development of the vaccine.

## 1. Introduction

Group B Streptococcus (GBS) or *Streptococcus agalactiae* is a beta-hemolytic, Gram-positive bacterium. Based on the specificity of its capsular polysaccharide (CPS), it can be classified into 10 serotypes: Ia, Ib, II, III, IV, V, VI, VII, VIII, and IX [1,2,3]. GBS can colonize women’s vaginas, intestines, and urethras during pregnancy. Thus, the newborns can be infected with GBS directly from their mothers when they pass through the maternal genital tracts during delivery. GBS causes a spectrum of diseases, including stillbirth, maternal infection, and early- and late-onset sepsis in newborns, and it may result in preterm delivery and hypoxic-ischemic encephalopathy through rectovaginal colonization [4].

Several guidelines have been published over the years to tackle GBS infections. For example, as early as 1996, the Centers for Disease Control and Prevention (CDC) and relevant professional societies developed guidelines for the prevention and management of perinatal group B streptococcal disease [5]. The consistent guidelines for the prevention and management of perinatal GBS disease have been released by the American Academy of Pediatrics (AAP) and the American College of Obstetricians and Gynecologists (ACOG) in 2019 and 2020, respectively [6]. Essential related guidelines and their important general principles are summarized in Table 1 [5,6,7,8,9]. With the implementation of these strategies, the number of children deaths has fallen globally. However, there has been less improvement in reducing neonatal mortality and stillbirth rates, with 2.7 million neonatal deaths and 2.6 million stillbirths in 2015 [10,11,12]. The global pooled incidence of invasive infantile GBS disease is estimated to be 0.49 (95% Cl 0.43–0.56) per 1000 live births, which can still be considered a high rate [2]. For the prevention of early-onset GBS disease, antibiotics such as penicillin, ampicillin, or cefazolin are currently available, but at present, there is no suitable method for the prevention of late-onset GBS disease [6]. In addition, antibiotic resistance has become a global issue. As this review discusses the key GBS virulence factors, the current available drug treatments are not discussed here [13,14].

More than twenty different virulence factors contribute to the GBS pathogenesis. These include adhesins, enzymes, carbohydrates, and other proteins. Most of the adhesins mediate GBS colonization in the epithelium of the vaginal tract, enabling its transmission to newborns as they pass through the genital tract at birth. Adhesins have also been associated with EOD, such as early-onset sepsis [15]. Other virulence factors, such as HvgA, hemolytic pigment, and alpha C protein, can facilitate GBS invasion of different tissues, such as the brain, placenta, and cervix, leading to a poorer physical condition [16,17,18,19]. In gaining access to the blood circulation, GBS can invade the brain epithelium, causing meningitis, one of the late onset diseases (LODs) in newborns, which is fatal [20].

This paper will discuss major characterized virulence factors and their involvement in the pathogenesis of GBS. A better understanding of these mechanisms has significant implications for overcoming the challenges associated with vaccine development and the development of new treatment regimens for GBS.

## 2. Virulence Factors Associated with Interaction with the Vagina

The vagina is thought to be the main reservoir of GBS [21]. GBS colonization in the vagina of pregnant women is a major risk to the newborn. These virulence factors are associated with dissemination, immune evasion, and damage to tissue, and allow GBS survival in the hostile vaginal environment.

### 2.1. Adhesins

The first step for GBS colonization of the vagina is the adhesion to its epithelial cells via surface-associated adhesins. Several adhesion factors enable GBS to bind to components of the extracellular matrix (ECM), thereby enhancing its ability to penetrate the host mucosal barrier and spread to other host tissues.

**Srr1/Srr2:** Srr1/Srr2 are structurally characterized by an extended N-terminal signal sequence, two highly glycosylated serine-rich repeats (Srr), at least one non-repeat binding region (BR), and a C-terminal cell wall anchoring domain carrying the LPxTG (Leu-Pro-x-Thr-Gly) pattern [22]. Srr is glycosylated by glycosyltransferases (Gtfs), which includes a two-protein glycosyltransferase complex (GtfAB), Nss, and Gly [23]. The large loci encoding Srr1 and Srr2 are located at different chromosomal positions with similar genetic organization, and the genes mediating Srr glycosylation are also encoded by these loci [24]. Furthermore, Srr1 and Srr2 are structurally similar but show only 32% sequence identity at the amino acid level [25]. Therefore, only limited homology is shown (<20% concordance) [26]. Genome-wide association studies have shown that highly pathogenic strains of *agalactiae* could almost exclusively produce Srr2 and Gtfs [26,27].

The GBS Srr family of glycoproteins as surface-associated fibrinogen binding proteins (Fbs) binds to a single tandem repeat region of human fibrinogen via a ‘lock, dock and latch’ mechanism [15]. This binding leads to a series of ordered conformational changes in Srr and results in enhanced adhesion to the target cells [15]. GBS strains can only express one of Srr1 and Srr2 [28]. Clonal complex (CC)-17 strains (capsular serotype III) are highly virulent producers of the Srr2 [15,26]. Srr2 is also associated with the CC-1 strain, which is one of the major CC strains in Bone and joint infections (BJI), while BJI is the second cause of invasive group B streptococcal (GBS) infection [29]. Most GBS clinical strains express Srr1, which binds to fibrinogen and keratin 4 and thereby mediates adhesion to the vaginal and cervical epithelium [15]. Interestingly, although Srr1 is the most dominant glycoprotein, it does not bind plasminogen and plasmin, whereas Srr2 can bind them, and it is the characteristic surface glycoprotein of ST-17 strains [25]. Moreover, Srr2 also binds with greater affinity to fibrinogen, further enhancing the adherence of highly pathogenic strains to the target tissue [15].

Several studies showed that deletion of the entire Srr1 glycoprotein, or of just the latching structural domain of Srr1, decreases vaginal colonization of GBS [28,30]. Although often asymptomatic after the colonization, this can lead to amnionitis or bacteremia in pregnant women [31]. Among the pathogenic mechanisms, Srr1 glycosylation is important [24]. Srr1 glycosylation enhances the stability of Srr1 by resisting protease inactivation, thereby prolonging adhesion and persistence. Compared to members of the RofA-like protein (RALP) family of transcriptional regulators responsible for the bacteria–host interaction, the proteins encoded by *Rga* showed high similarity to these members [32]. The *Rga* gene is particularly important for Srr1, and deletion of the *Rga* gene results in a significant reduction in Srr-1 expression. However, strains carrying Srr2 do not encode a *Rga* equivalent [24,33]. Therefore, subsequent in-depth studies of *Rga* in *S. agalactiae* may provide new strategies for disease treatments.

**FbsA, FbsB, FbsC:** FbsA, FbsB, and FbsC are three members of the family of fibrinogen-binding proteins encoded by GBS. They adhere to human epithelial cells to promote vaginal colonization [34]. FbsA and FbsB were most frequently detected in infected pregnant women [35]. Meanwhile, not only is FbsB found in some of the major CCs in BJI strains, the CC10 and CC23, but it also promotes the epithelial invasion of GBS [29,36]. FbsC, which contains two bacterial immunoglobulin-like tandem repeat domains and a C-terminal cell-wall-anchoring motif (LPxTG), can also mediate biofilm formation, which can promote GBS colonization of the vagina. However, FbsC was not expressed in those clinical GBS isolates belonging to the highly pathogenic lineage ST17. In addition, it has been shown that FbsC can help GBS to colonize the brain with immunoprotective activity [34].

**PbsP:** The plasminogen binding surface protein (PbsP) is a surface protein with crucial functions in GBS pathophysiology. This adhesion protein has two streptococcal surface repeat structural domains, a methionine- and lysine-rich region, and an LPxTG cell wall anchoring pattern, mediating plasminogen binding to enhance the ability to colonize and invade host tissues [37]. A two-component system (TCS) homologous to the staphylococcal virulence regulator SaeRS is upregulated in vivo during GBS infection of the vagina, and PbsP is one of the targets of SaeRS. By employing a mouse model of vaginal carriage and using transcriptome sequencing (RNA-Seq), it was found that SaeRS promotes the expression of PbsP by using a component of vaginal irrigation fluid, which is still unknown as a signal to play an important regulatory role in PbsP [38]. A recent study discovered that the specific sequences required for this interaction are in the methionine and lysine-rich (MK-rich) domain of PbsP and the Kringle 4 lysine-binding sites (LBS) of plasminogen. They also found that the presence of lysine or other positively charged amino acid residues in the MK-enriched region is not necessary for the region to bind to plasminogen [39]. The finding of the MK-rich domain and Kringle 4 LBS may provide a new direction for vaccine development. In the pathogenesis of invasive infections, the binding of microbial pathogens to the host vitronectin (Vtn) is prevalent. In another study targeting Vtn, the pretreatment of cells with anti-Vtn antibodies or Vtn resulted in the inhibition and promotion of GBS adhesion and invasion of epithelial cells, respectively. Thus, Vtn was further found to act as a bridge between the streptococcal surface repeat (SSURE) domain of PbsP on the surface of GBS and the host integrin. In addition, inhibition of the interaction between PbsP and extracellular matrix components prevents GBS colonization [40].

**Pili:** The pili are GBS-encoded cell-wall-anchored appendages that protrude from the bacterial surface and each gene encodes three structural subunit proteins: capillary axis backbone protein (PilB), capillary tip (PilA, capillary-associated adhesion protein binding collagen type 1), and capillary base (PilC) [21], bearing a C-terminal LPxTG motif and two subfamily C sortases (SrtC) involved in covalent polymerization of the subunits [41,42,43]. GBS strains also possess the conserved “housekeeping” sorting enzyme A (SrtA), which is involved in the covalent assembly of pili on the cell wall. According to research, the deletion of SrtA does not affect the polymerization of a pilus but results in a reduction in pilus expression on the cell surface. In one model of GBS pilus assembly, the polymerization of pilus structures is handled by pilus island (PI) sortase, and SrtA utilizes the accessory pilus subunit GBS150 as an anchoring protein to covalently attach it to the cell wall [43].

In addition to the capsule, the pilus has now been identified as an essential factor in increasing the pathogenicity of GBS. Genome mining identified two types of PI in GBS, PI-1 and PI-2, with PI-2 having two alleles, PI-2a and PI-2b [44]. Furthermore, genes for synthesis and assembly of the pilus are included in the PI [43,45]. Thus, one study testing 57 GBS cases from pregnant women by multiplex PCR for the determination of PI-1, PI-2a, and PI-2b found that most GBS contained PI-1+PI-2a and the presence of these pili stabilizes colonization [46].

**Lmb:** The laminin binding protein (Lmb) mediates the attachment of GBS to human laminin, which is crucial for bacterial colonization and invasion [47]. According to recent research [48], GBS is less invasive in the *lmb* mutant strains, especially in human brain microvascular endothelial cells (HBMECs). During infections, neutrophils release large amounts of calprotectin, which binds zinc efficiently, thereby inhibiting bacterial growth [48]. Interestingly, a recent study has shown that the *lmb* gene in GBS promotes resistance to the reduction in zinc caused by calprotectin. As GBS alters its own zinc transport mechanisms, upregulating genes encode zinc-binding proteins, lmb, adcA, and adcAII, thus assisting GBS to bind zinc. In addition, mice infected with the Δ*lmb* mutant not only had less GBS in their brains, but also had decreased mortality [49]. This further demonstrates the importance of the virulence of lmb to GBS.

**ScpB:** ScpB or C5a peptidase is a surface-associated serine protease. It not only prevents complement activation by cleaving C5a, a neutrophil chelator, but also helps bacteria bind fibronectin [50,51]. Through this fibronectin binding, it could aid in the GBS invasion of human epithelial cells [50]. In 90 isolates collected from invasive and non-invasive isolates from adults, serotype III accounted for 68.9% of all 90 isolates, and ScpB was detected in all 90 isolates [52]. In addition, analysis of 242 GBS strains, including 95 colonizers without pathogenicity and 68 pathogenic strains isolated from pregnant women, and 79 strains isolated from newborns with sepsis, revealed that the frequency of ScpB was significantly higher in neonatal strains. This shows that ScpB can be very well used to help GBS infect newborns with sepsis [53]. However, when ScpB is cross-linked to fibrin mediated by factor XIIIA (FXIIIA), GBS entrapment in the fibrin clot increases, allowing for reduced transmission of systemic infections [54]. This may provide new insights for future GBS treatments.

**HylB:** HylB denotes hyaluronidase or hyaluronate lyases, an exolytic enzyme released by the GBS. HylB can cleave the high-molecular-weight glycosaminoglycan polymer of hyaluronic acid, which serves as the epithelial extracellular matrix component. HylB cleavage of hyaluronic acid breaks the maternal–fetal barrier and enables the travel of GBS from the vagina to the fetus to cause fatal infection and damage [55]. The byproducts, such as disaccharide fragments produced from the cleavage of hyaluronic acid (HA), can bind to Toll-like receptors 2 and 4, blocking the pro-inflammatory cascade response induced by some GBS components [56]. The blocked TLR2- or TLR4-mediated response leads to immunosuppression, making the ascending infection possible. However, the non-hyaluronidase mutant GBS was found to be cleared by immune responses as they lack the HylB enzyme [55].

The non-pigmented and non-hemolytic GBS strains both show their high infectivity by increasing the activity of the hyaluronidase HylB [57]. It is found that HylB as a potent virulence factor allows GBS to circumvent neutrophil responses, invade the amniotic cavity, and cause fetal bacteremia and preterm labor in a unique non-human primate model used to mimic human pregnancy and inoculated with hyaluronidase (HylB)-expressing GBS [58]. HylB-proficient GBS confers more resistance to neutrophils by inhibiting the production of reactive oxygen species (ROS). Neutrophils produce ROS via Toll-like receptors (TLRs)-2/4 signaling, and pro-inflammatory HA can participate in this signaling, thereby promoting ROS production, whereas HylB can block ROS production by cleaving HA [58]. The results suggest HylB as a potential therapeutic target for invasive GBS disease during pregnancy. Identification of a HylB-specific inhibitor could lead to new treatments for GBS and have far-reaching impacts on maternal and neonatal health worldwide.

### 2.2. Hemolytic Pigment

Hemolytic activity in GBS is due to the ornithine rhamnolipid pigment (hereafter called “hemolytic pigment” or “pigment”, also known as Granadaene) [18,59], which is produced by the genes of the *cyl* operon [60,61]. Surprisingly, it has been shown that Granadaene can also be produced by Lactobacillus heterologously expressing the GBS *cyl* operon, and that the Granadaene produced is functional [62]. Among the *cyl* operon, the *cylE* gene is necessary for pigment production [18,60,61], and transcription of *cyl* genes is negatively regulated by the CovR/S two-component system (also known as CovR/S TCS) [18]. As a major virulence factor in GBS, Granadaene not only has pigmentary and hemolytic activity but also effectively resists the elimination by mast cells, macrophages, and neutrophils [18,19]. Additionally, it can penetrate the human placenta [18], invading the amniotic cavity and causing severe fetal damage [63]. However, the hemolytic activity of GBS can be inhibited by photobleaching of granadaene and the sensitivity of GBS to reactive oxygen species, such as hydrogen peroxide, leading to increased membrane permeability, thus reducing its activity [64].

Additional research investigated the effect of hemolytic pigment on platelets [65]. By using the pigmented LUMC16 strain and its non-pigmented isogenic LUMC16Δ*cyl*X-K mutant, it was observed that platelets responded to stimulation with the LUMC16 strain but not to the unpigmented LUMC16Δ*cyl*X-K strain within 30 min. Thus, GBS pigment can induce initial platelet activation. Further observations revealed a gradual decrease in platelet viability with increasing time of infection in the LUMC16 strain and that platelets were killed when the pigment concentration increased to a certain level (1 and 2 μM). In contrast, the cells remained viable when infected with non-pigmented bacteria. That is, therefore, a good indication that pigments can cause platelet death [65]. Although some advancements have been made, the structural features responsible for hemolysis and cytotoxicity are still to be investigated [62].

## 3. Virulence Factors Associated with Interaction with the Cervix

The cervix controls and restricts the entry of microorganisms into the uterus by secreting mucus, cytokines, and antibacterial peptides. These secretions are not only effective in the prevention of infections but are also protective to the fetus developing in the uterus. If this barrier is breached, bacteria may enter the uterine cavity and cause premature birth [60].

### 3.1. Alpha C Protein

The alpha C protein (ACP) is the prototype of a family of Gram-positive bacterial surface proteins. It facilitates the entry of GBS into human cervical epithelial cells and traverses the cell layer by binding to glycosaminoglycans (GAG) on the surface of the host cell [66]. The prototype alpha C protein of GBS from strain Ia/C A909 includes a series of nine identical 246 bp tandem repeat units. It was found that deficiency in the tandem repeat region of the ACP occurs through a recA-independent recombination pathway and affects the protective potency and immunogenicity of the protein, so these deficiencies may be a mechanism of virulence in GBS [67].

### 3.2. Capsule

GBS capsules consist mainly of carbohydrates with the capsule polysaccharide synthesis (CPS) operon driving its synthesis. The product of the *cpsE* gene in this operon is crucial for biofilm formation [68]. Recent studies demonstrated that, compared to the wild-type strain, the GBS *cpsE* mutant secretes fewer carbohydrates, resulting in weak capsules, and, therefore, a reduced growth of biofilms [68]. Based on the capsular antigen, GBS has been classified into ten serotypes (Ia, Ib, and II-IX) as of today. Type Ib (44.4%) was the most common type of capsule, and the next most common types were Type III (40.7%), Type II (11.1%), and Type Ia (3.7%) [69].

Streptococcal polysaccharide capsules defend bacterial cells from deposition of complement, opsonization, and phagocytosis [70,71]. Capsules contain the α2,3-linked sialic acid (Sia) residues, which are analogs to a human cell surface glycocomplex epitope [72]. Typically, platelets are degranulated by contacting microorganisms through chemotaxis and released kinins and small cationic platelet microbicidal proteins (PMPs). However, the direct contribution of platelets to the killing of GBS has not been described [73]. Studies have shown that CPS Sia can effectively inhibit the killing of GBS by human platelets and resist platelet-derived antimicrobial components. GBS without Sia expression have increased susceptibility to thrombin-activated platelet release [74,75]. Meanwhile, CPS Sia can also bind to Ig superfamily lectins (Siglecs) to inhibit the activation of neutrophils and macrophages [76]. Numerous studies have demonstrated that GBS can bind to the human platelet surface receptor Siglec-9 in a Sia-dependent manner [77,78], thereby inhibiting platelet activation. A recent study demonstrated that eliminating the inhibitory Siglec-E in a mouse model effectively abolished the inhibitory effect of GBS on platelet activity [68]. It follows that CPS Sia can not only inhibit platelet activation by interacting with inhibitory Siglecs but also directly impact intrinsic resistance to platelets.

## 4. Virulence Factors Associated with Interaction with the Endometrium

The endometrium is the inner lining of the uterus containing crucial glands for menstruation and embryo implantation [79]. Although specific virulence factors that interact with the endometrium have not been identified yet, GBS invasion could be fatal to the fetus during gestation [2,4].

## 5. Virulence Factors Associated with Interaction with the Feto–Maternal Interface

The feto–maternal interface includes decidual stromal cells (DSCs), cytotrophoblasts (CTBs), and macrophages (Mφs), which are necessary for the protection of the fetus. In particular, CTBs can secret factors to regulate immune cells at the feto–maternal interface such as factors inhibiting GBS-stimulated Mφ NFκB activity, tumor necrosis factor α (TNFα), and Matrix metallopeptidase 9 (MMP9) production [80]. However, GBS has developed stronger virulence factors to resist host defense and promote its survival.

### Membrane Vesicles (MVs)

Among the numerous causes of preterm birth, infection of the fetal membrane (amniotic and chorionic) with extracellular membrane vesicles (MVs) cannot be ignored. The GBS MVs contain nucleic acids, lipids, and virulence factors such as hyaluronate lyases, sialidases, and C5a peptidase [81,82]. Moreover, the production and composition of GBS MVs are strain-dependent, with specific lineage functions related to virulence [81]. Placental membranes enclose the fetus and play an important role in its protection. Recent studies have found that GBS MVs have a strong influence on choriodecidual membranes and that their secretion of various virulence factors can diminish the integrity of the membranes [82]. Surve et al. described that the protease activity present in MVs led to the degradation of collagen [82]. The fetal membranes are located on a collagen basement membrane, which is composed of collagen types II and IV. Beneath this membrane, there is a fibrous layer containing collagen types I, III, V, and VI. Therefore, the main structural strength of the membrane is provided by collagen, and its degradation can result in the loss of membrane integrity.

Additionally, GBS MVs can lead to infiltration of neutrophils and lymphocytes under the epithelium of the amnion, thus promoting extensive inflammation in the chorio-decidua [82]. Although the mechanism remains unclear, it can cause apoptosis in the chorio-decidual membrane [82].

Reactive oxygen species (ROS) produced by recruited neutrophils are essential for host clearance of GBS [63], and it has been demonstrated that it is likely that GBS destroys the ROS through the release of hemolytic pigments from MVs, thereby weakening the defense of hosts against GBS and allowing the bacteria to survive [83]. Meanwhile, hemolytic pigments in MVs can also inhibit GBS killing by H_2_O_2_, a major component of the oxidative burst. Furthermore, lung observations in mice treated with non-hemolytic (NH) and hyperhemolytic (HH) MVs also revealed that hemolytic MVs may exacerbate lung injury and/or bacteremia in neonates infected with NH GBS [83]. These, therefore, suggest that hemolytic GBS MVs could contribute to the pathogenesis of GBS and neonatal morbidity and mortality.

## 6. Effects on the Newborn

Neonates can be infected with GBS in several ways, such as exposure to vaginal secretions at birth or through breast milk [84,85]. Invasive GBS disease is usually classified as EOD and LOD depending on when the newborn is infected with GBS after birth [20]. The infection occurring within 0 to 6 days after birth is termed EOD and 7 to 89 days after birth is termed LOD [86]. Sepsis accounts for the largest proportion of both EOD and LOD, but the health status of the newborns is worse in LOD, as meningitis often develops from sepsis and is a serious health risk for newborns.

### 6.1. HvgA

Hypervirulent GBS adhesin (HvgA), as an ST-17-specific surface-anchored protein, can enhance the GBC across the intestinal and blood–brain barriers. The bacteria, mediated by it, can spread into the bloodstream and central nervous system, leading to the development of meningitis [17]. In addition, it is confirmed that it plays an important role in triggering meningitis [16]. The strains isolated from pregnant women were found to encode surface protein virulence factors including HvgA, which helps it to acquire high virulence [87].

Studies have shown that serotype III is the most prevalent of the GBS serotypes and that this serotype accounts for most of the invasive neonatal disease (IND) cases, particularly LOD [88,89]. CC-17, the most common CC, has isolates that are almost exclusively serotype III. Moreover, according to research, CC-17 accounts for almost twice as many invasive isolates as non-invasive ones, and CC-17 is more common in patients with LOD than in those with EOD [90]. The reason for its greater occurrence in LOD patients is that the invasive nature of CC-17 allows it to further invade the brain through circulation, leading to meningitis, but this takes more time. The non-invasive GBS, on the other hand, mostly colonizes the maternal vagina and cervix and enters the fetus directly during birth through the birth canal, causing the newborn to become ill more quickly.

### 6.2. CAMP Factor

The CAMP factor, named after its discoverers Christie, Atkins, and Munch-Petersen [91], is a pore-forming toxin that synergistically interacts with the beta-toxin of *Staphylococcus aureus* to produce a co-hemolytic response that contributes to the identification of GBS in the clinical laboratory [38]. According to an earlier mouse experiment, the CAMP factor was thought to possibly bind to the Fc site of many mammalian Igs, leading to septicemia in mice and increasing the pathogenicity of GBS [92]. However, testing the contribution of CAMP factors with GBS strain COH1 and its homologous CAMP-deficient mutant (Δcfb) revealed no evidence that CAMP factors are required for GBS biofilm formation, and that CAMP factors do not contribute to GBS adhesion, invasion of VK2 cells, or expression of cytotoxicity in the presence or absence of *Staphylococcus aureus*. In addition, there was no difference in vaginal burden or persistence between COH1 and Δcfb strains in the mouse colonization model. Therefore, it is not likely that the biological targets for controlling maternal GBS colonization include CAMP factors [93]. Other data also suggest that the CAMP factor is not essential for systemic virulence of GBS [94]. A summary of the main GBS virulence factors is presented in Table 2 and Figure 1.

## 7. GBS Vaccine

Using a vaccine to prevent and address the serious effects of GBS on pregnant women and newborns is a good strategy, but vaccine development is far from as straightforward and convenient as it may seem. In April 2016, the WHO invited representatives from the industry, academia, public health agencies, and funding agencies to a consultation on GBS vaccines for the development of maternal immunization vaccines in low- and middle-income countries [99]. Based on the previous description of virulence factors, pili, hemolytic pigment, CPS, ACP, FbsC, and PbsP are promising targets for vaccine development. The following is a brief update on the latest progress in vaccine development for CPS and ACP.

**CPS:** GBS vaccines have been in various stages of development in recent decades and, currently, a multivalent capsule polysaccharide (CPS) conjugate vaccine is in clinical trials [100]. The production of this vaccine requires a polysaccharide–protein conjugate, and the complex is obtained by chemically cross-linking the CPS extracted from the purified capsules of GBS cells with the carrier protein [101]. It is hoped that this vaccine will be introduced into the clinic as soon as possible to provide better protection for pregnant women.

**ACP:** Thanks to a comprehensive understanding of ACP, a new vaccine was developed. The GBS-NN vaccine was designed based on the GBS Alpha C and Rib proteins and was previously only safe to use in healthy, non-pregnant women [55]. To understand the effectiveness of the vaccine in pregnant women, a mouse model that reproduces the outcome of human GBS infection was used to demonstrate that the GBS-NN vaccine resulted in higher GBS-NN-specific IgG titers in vaccinated mice. Although the vaccine did not eliminate GBS infection in the ascending stages in pregnancy, it could reduce fetal mortality in utero and allowed the newborn to acquire more protective antibodies from the mother [102].

Although the development of new vaccines can be challenging, it is effective in reducing the serious damage caused by GBS. To help overcome the GBS infection earlier, more effort and funding should be paid in the future. The main GBS vaccines under clinical trial are summarized in Table 3.

## 8. Conclusions

Group B Streptococcus (or GBS) can colonize the body asymptomatically and cause severe diseases, such as sepsis and meningitis. Pregnant women and newborns are common targets. Although several measures have been implemented in response to GBS infections, the global burden of GBS disease remains high. GBS evolved a set of virulence factors, including adhesins and hemolytic pigments, to promote its ability to colonize and invade human hosts. While there has been a great deal of research in recent years to improve our understanding of GBS, more exploration is still needed on the structure of some of the pathogenic factors of GBS and how it survives as both a symbiont and a pathogen. Filling this knowledge gap could be crucial for the development of future treatment options. In the meantime, only a full understanding of the pathogenic mechanisms of GBS can lead to more effective prevention of GBS disease.

## Figures and Tables

**Figure 1 microorganisms-10-02483-f001:**
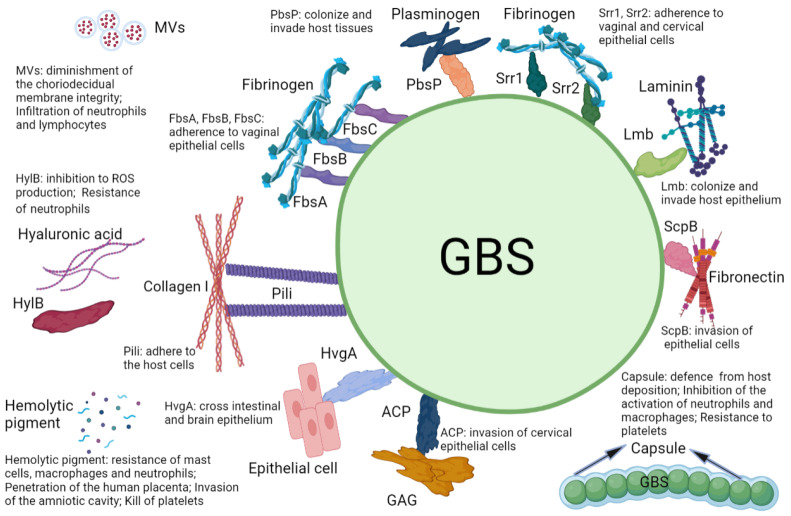
Summary of GBS virulence factors elucidated in this review, with their specific targets and mechanisms.

**Table 1 microorganisms-10-02483-t001:** Summary of the key guidelines and their general principles for prevention of GBS disease.

Organization	Year	Guidelines	Important General Principles
CDC	1996	Prevention of perinatal group B streptococcal disease: a public health perspective.	Using one of two prevention strategies is recommended (details in [7]).
CDC, AAP, and ACOG	1996	Recommendations for intrapartum prophylaxis to prevent perinatal GBS disease (consensus guidelines)	To prevent early onset disease (EOD), intrapartum antibiotic prophylaxis (IAP) is recommended for pregnant women using antenatal cultures or a risk-factor-based approach.
CDC	2002	Prevention of perinatal group B streptococcal disease: revised guidelines from CDC.	Universal culture-based screening is recommended for all pregnant women at 35–37 weeks’ gestation to identify women who require intrapartum antibiotic prophylaxis (IAP), as well as a neonatal management algorithm for secondary prevention of GBS EOD
CDC	2010	Prevention of perinatal group B streptococcal disease: 2010 revision	Selection of specific antibiotics for the use of IAP in women in preterm labor and women with premature rupture of membranes.
AAP	2019	Prevention and management of perinatal GBS disease	Recommend that infants less than, greater than, or equal to 35 weeks’ gestation should be considered separately when performing risk assessment for GBS.
ACOG	2020	Prevention and management of perinatal GBS disease	Adjusting the start of antenatal antibiotic prophylaxis for pregnant women from 36 0/7 to 37 6/7 weeks of gestation.

**Table 2 microorganisms-10-02483-t002:** Summary of GBS virulence factors covered in this review, with their prevalence, specific targets, and mechanisms.

Virulence Factor	Specific Target	Mechanism	Prevalence	Reference
Serine-rich repeats1 (Srr1)	Fibrinogen	Adherence to vaginal epithelial cells, cervical epithelial cells	Common	[15,25,89]
Serine-rich repeats2 (Srr2)	Fibrinogen, Plasmin, Plasminogen	Adherence to vaginal epithelial cells, cervical epithelial cells	Common	[15,25,89]
Fibrinogen-binding proteins (FbsA, FbsB, FbsC)	Fibrinogen	Adherence to vaginal epithelial cells	Common	[34,89]
Hypervirulent GBS adhesin (HvgA)	Unknown	Cross intestinal epithelium, brain epithelium	Common	[16,17,89]
Plasminogen binding surface protein (PbsP)	Plasminogen	Colonize and invade host tissues	Unknown	[37]
Pili	Collagen I	Adherence to host cells	Common	[28,43,89,95,96]
Laminin binding protein (Lmb)	Laminin	Colonize and invade host epithelium	Common	[47,89,97]
C5a peptidase (ScpB)	C5a, Fibronectin	Invasion of epithelial cells	Common	[50,51,89]
GBS hyaluronidase (HylB)	Hyaluronic acid	Inhibition to ROS production; resistance to neutrophils	Common	[56,58,98]
Hemolytic pigment	Neutrophils, Mast cells, Macrophages, Platelets	Resistance to mast cells, macrophages, and neutrophils; penetration of the human placenta; invasion of the amniotic cavity; kills platelets	Common	[18,19,63,65]
Alpha C protein (ACP)	Glycosaminog-lycans (GAG)	Invasion of cervical epithelial cells	Unknown	[66]
Capsule	Siglecs	Defense from host deposition; inhibition of the activation of neutrophils and macrophages; resistance to platelets	Common	[70,71,74,75,76]
Membrane vesicles (MVs)	Collagen	Diminishment of the choriodecidual membrane integrity; infiltration of neutrophils and lymphocytes	Common	[81,82]

**Table 3 microorganisms-10-02483-t003:** Summary of GBS vaccines in clinical trials.

Vaccine Candidate	Basic Components	Clinical Trial	Reference
Native polysaccharide vaccines	Capsule polysaccharide (CPS)	Ineffective	[103]
GBS glycoconjugate vaccine	Capsule polysaccharide (CPS)	Phase I	[104,105]
Trivalent (Ia, Ib, and III) CRM197 conjugate vaccine	Capsule polysaccharide (CPS)	Phase Ib/II	[106]
Pentavalent conjugate vaccine (Ia, Ib, II, III, and V)	Capsule polysaccharide (CPS)	Phase I/II	[107]
Hexavalent vaccines (Ia, Ib, II, III, IV, V)	Capsule polysaccharide (CPS)	Phase I/II	[99]
Prototype recombinant alpha-like protein subunit vaccine (GBS-NN)	Highly immunogenic N-terminal domains of Alpha C and Rib (GBS-NN)	Phase I	[99,108,109]

## Data Availability

Not applicable.

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
