# Peer review of "Group B Streptococcus: Virulence Factors and Pathogenic Mechanism"

_microorganisms, 2022, doi:10.3390/microorganisms10122483_

Round 1
Reviewer 1 Report
The manuscript is a review paper on the virulence factors of the pathogenic Group B Streptococcus. Knowledge of the virulence factors and their regulations is important for finding new strategies for treatment of GBS. However, the manuscript is written with many confuse sentences, which make the content not accurate. Also, many wrong terminologies are used, resulting in a doubt on the accuracy of the content. The manuscript needs to be more concise and focused. The manuscript has two tables, but lacks illustrations. The manuscript lacks updated references. There are many jumps in the text, which should be more structured. (For instance, sections 2.6, 2.7. 2.9 are misplaced).
General comments:
A space should be made before the reference numbers.
Bacterial names should be in italics.
I would highly recommend preparing an illustration of the different virulence factors and their regulation and involvement in GBS virulence.
Please double check that everything written is accurate and precise. There are several places with doubts.
Please cite the original paper demonstrating the specified issues.
Specific comments:
Line 14: I would suggest writing: "is important" instead of "will be important".
Line 29: Write the full name of CDC.
Introduction: The authors describe a long paragraph of the existence of guidelines for the prevention and management of perinatal group B streptococcal diseases. It would be proper to add a Table that outline the general principles of these guidelines.
Line 48: Too many times "like". I would suggest writing "such as".
Line 51: Please remove: "with assistance several days later".
Line 52: Define LODs.
Lines 53-54: I would suggest rephrasing the sentence to: "This paper will discuss major characterized virulence factors and their involvement in the pathogenesis of GBS"
Line 56: I would remove "virulence". ("treatment regimens for GBS").
The sentence: "The factors will be discussed." in lines 61-62 can be removed.
Please check the correctness of the references. For instance, the "glycosylation islands" are not described in reference 19. What do you mean with "glycosylation islands"? These can be illustrated in a Figure.
Line 72: Please define SRRP.
Line 74: "Whereas" can be removed.
The following sentence should be explained: "Srr2-associated glycosyltransferases (GTs) are more specific." Explain what do you mean with "associated" and what is the specificity of Srr1 versus Srr2. The further text is a little bit confusing especially when the authors write that Srr2 is a homolog to Srr1. What are the differences between these two components?
Line 82: Define CC1.
Line 86: Maybe you mean: "although Srr1 is the most dominant,"
It is not clear from the text whether Srr1 and Srr2 are expressed on the same bacteria, or they are expressed on different strains?
The enzymes involved in Srr glycosylation should be mentioned.
Some more words should be dedicated to the Rga transcription factor.
Decide either to write LPXTG or LPxTG (Leu-Pro-x-Thr-Gly)
Line 117: Describe "the component".
Line 119: Define MK.
Line 120: Define LBS.
Line 122: Explain how the positively charged amino acids might be a target of vaccine development if they are not important for the binding to plasminogen.
The abbreviation Plg was used at the second time plasminogen is mentioned.
Line 123: Vtn is vitronectin and not Vitamin C.
What do you mean with: " that started with vitamin C"?
The sentence in lines 122-126 is long and not clear. Please rephrase.
Lines 126-127: Remove: "This further demonstrates that".
The structure of the GBS pili should be illustrated. Please double check if the names of the PilA-C provided are the right concepts. For instance, the names: capillary axis backbone protein (PilB), capillary tip (PilA, capillary-associated adhesion protein binding collagen type 1) and capillary base (PilC). Please provide a reference that has used these concepts.
Define PI sortase.
The pilus islands should be illustrated.
Line 144: I think you meant: "these pili"
Line 149: The following sentence can be removed as it is out of context: " However, it does not affect the release of IL-8 from HBMEC".
How can Lmb prevent the calprotectin binding to zinc?
Line 158: Instead of "splitting", I would suggest writing "cleaving".
C5a is not a neutrophil chelator, but a potent neutrophil chemoattractant.
Line 161: Incomplete sentence: " serotype III accounting for 68.88%" Is it the percentage of invasive or non-invasive isolates or both?
Line 166: Please define FXIIIA.
Line 178: Correct to: " lack the HylB enzyme."
How does HylB prevent ROS production by neutrophils?
Line 243: Explain more about the mechanisms how platelets can kill GBS. And how CPS Sia inhibits synthetic platelet-associated antimicrobial peptide. In the text: once you write CPS Sia, then GBS Sia. Please be consequent and use the right terminology.
Line 299 is a repetition of a previous section and should be transferred to the part describing EOD and LOD. Lines 311-320 should also be described in this part.
Table 1 is incomplete and should be more concise.
Lines 370-371: ?? 6 Pa??

Author Response
There are many jumps in the text, which should be more structured. (For instance, sections 2.6, 2.7. 2.9 are misplaced).
Response: Thank you for your great advice. There is a misunderstanding with my titles of sections. Titles of sections 2.3-2.11 were changed.
General comments:
- A space should be made before the reference numbers.
Response: Thank you for your good advice. A space was made before the reference numbers.
- Bacterial names should be in italics.
Response: Thanks for your essential advice. All the bacterial species names throughout the manuscript were italicized.
- I would highly recommend preparing an illustration of the different virulence factors and their regulation and involvement in GBS virulence.
Response: Thank you for your recommendation. It is really a good idea. I made an illustration of the different virulence factors and their regulation and involvement in GBS virulence. Hope it will meet your requirements.
- Please double check that everything written is accurate and precise. There are several places with doubts.
Response: Thanks for your good advice. The manuscript was double checked. It is more accurate and precise now.
- Please cite the original paper demonstrating the specified issues.
Response: Thank you for your important suggestion. There were mistakes with reference before and some references were changed to original paper.
Specific comments:
- Line 14: I would suggest writing: "is important" instead of "will be important".
Response: Thank you for your good comment. "will be important" was replaced by "is important" in line 14.
- Line 29: Write the full name of CDC.
Response: Thank you for your good advice. Full name of CDC was written.
- Introduction: The authors describe a long paragraph of the existence of guidelines for the prevention and management of perinatal group B streptococcal diseases. It would be proper to add a Table that outline the general principles of these guidelines.
Response: Thank you for your great advice. This makes my manuscript fuller. Table 1 was added that outline the general principles of these guidelines.
- Line 48: Too many times "like". I would suggest writing "such as".
Response: Thank you for your good advice. Each word "like" was replaced by "such as" in different expression .
- Line 51: Please remove: "with assistance several days later".
Response: Thanks for your suggestion."with assistance several days later" was removed.
- Line 52: Define LODs.
Response: Thank you for your good advice. "LODs" was defined in Line 52.
12.Lines 53-54: I would suggest rephrasing the sentence to: "This paper will discuss major characterized virulence factors and their involvement in the pathogenesis of GBS"
Response: Thanks for your good tip. The sentence was rephrased to "This paper will discuss major characterized virulence factors and their involvement in the pathogenesis of GBS".
13.Line 56: I would remove "virulence". ("treatment regimens for GBS").
Response: Thank you for your great advice. "virulence" was removed.
- The sentence: "The factors will be discussed." in lines 61-62 can be removed.
Response: This is a good advice."The factors will be discussed." in lines 61-62 was removed. Thank you.
15.Please check the correctness of the references. For instance, the "glycosylation islands" are not described in reference 19. What do you mean with "glycosylation islands"? These can be illustrated in a Figure.
Response: Thanks for your good advice.“Putative genomic islands, known as glycosylation islands, are responsible for encoding Srr proteins. And glycosylation islands carried by Streptococcus agalactiae strains can encode either Srr1 or Srr2, with the Srr1 island having five more accessory genes than the Srr2 island” was deleted.
16.Line 72: Please define SRRP.
Response: Thanks for your great advice. I have changed SRRP to Srr proteins, as SRRP is a wrong abbreviation.
- Line 74: "Whereas" can be removed.
Response: Thanks for your good advice. "Whereas" was removed.
- The following sentence should be explained: "Srr2-associated glycosyltransferases (GTs) are more specific." Explain what do you mean with "associated" and what is the specificity of Srr1 versus Srr2.
Response: Thank you for your good suggestion. "Srr2-associated glycosyltransferases (GTs) are more specific." was changed to “Srr is glycosylated by glycosyltransferases (Gtfs), which includes a two-protein glycosyltransferase complex (GtfAB), Nss, and Gly”. Since "Srr2-associated glycosyltransferases (GTs) are more specific." is not accurate.
- The further text is a little bit confusing especially when the authors write that Srr2 is a homolog to Srr1. What are the differences between these two components?
Response: This is a grest question, thanks. I have added “ The large loci encoding Srr1 and Srr2 are located at different chromosomal positions with similar genetic organization. The genes mediating Srr glycosylation are also encoded by these loci. Srr1 and Srr2 are structurally similar but show only 32% sequence identity at the amino acid level. Therefore, only limited homology is shown (<20% concordance)”. Hope this can answer your question.
- Line 82: Define CC1.
Response: Thanks for your good advice. CC1 was defined: Clonal complex (CC) 1.
- Line 86: Maybe you mean: "although Srr1 is the most dominant,"
It is not clear from the text whether Srr1 and Srr2 are expressed on the same bacteria, or they are expressed on different strains?
The enzymes involved in Srr glycosylation should be mentioned.
Some more words should be dedicated to the Rga transcription factor.
Decide either to write LPXTG or LPxTG (Leu-Pro-x-Thr-Gly)
Response: Thank you for your good question and advice. Changed to "although Srr1 is the most dominant,". Whether Srr1 and Srr2 are expressed on the same bacteria, or they are expressed on different strains was explained. The enzymes involved in Srr glycosylation was mentioned.Decided to write LPxTG (Leu-Pro-x-Thr-Gly). Some more words were dedicated to the Rga gene.
- Line 117: Describe "the component".
Response: Thank you for your great suggestion . Since "the component"is still unknown, I added “ it is still unknown” in the manuscript.
- Line 119: Define MK.
Response: Thank you for your important advice. MK was defined: methionine and lysine-rich (MK-rich).
- Line 120: Define LBS.
Response: Thanks for your good advice. LBS was defined: Kringle 4 lysine-binding sites (LBS).
- Line 122: Explain how the positively charged amino acids might be a target of vaccine development if they are not important for the binding to plasminogen. The abbreviation Plg was used at the second time plasminogen is mentioned.
Response: These suggestions are very important, thanks. The sentence was changed to “MK-rich domain and Kringle 4 LBS may provide a new direction for vaccine development”. The abbreviation Plg was removed to the first time plasminogen is mentioned.
- Line 123: Vtn is vitronectin and not Vitamin C.
What do you mean with: " that started with vitamin C"?
Response: Thank you for your good discovery. I wrote the wrong word and“Vitamin C”was changed to vitronectin.
28.The sentence in lines 122-126 is long and not clear. Please rephrase.
Response: Thanks for your good advice. The sentence in lines 122-126 was rephrased to clearer sentences.
29.Lines 126-127: Remove: "This further demonstrates that".
Response: Thank you for the correction. "This further demonstrates that" was removed.
- The structure of the GBS pili should be illustrated. Please double check if the names of the PilA-C provided are the right concepts. For instance, the names: capillary axis backbone protein (PilB), capillary tip (PilA, capillary-associated adhesion protein binding collagen type 1) and capillary base (PilC). Please provide a reference that has used these concepts.
Define PI sortase.
The pilus islands should be illustrated.
Response: This is a really important advice. PI sortase was defined:pilus island (PI) sortase. A reference was provided for these concepts:Armistead, B., et al., The Double Life of Group B Streptococcus: Asymptomatic Colonizer and Potent Pathogen. J Mol Biol, 2019. 431(16): p. 2914-2931. However, I only found this article using PilA-C. Pilus islands were illustrated with “Genome mining identified two types of PI in GBS, PI-1 and PI-2, with PI-2 having two alleles, PI-2a and PI-2b. Furthermore, Genes for synthesis and assembly of the pilus are included in the PI.”
- Line 144: I think you meant: "these pili"
Response: What you noted is right. I used "these pili" in Line 144. Thank you.
- Line 149: The following sentence can be removed as it is out of context: " However, it does not affect the release of IL-8 from HBMEC".
Response: Thank you for your good advice." However, it does not affect the release of IL-8 from HBMEC" was removed.
33.How can Lmb prevent the calprotectin binding to zinc?
Response: This is a good question. Added more information: Interestingly, a recent study has shown that the lmb gene in GBS can help it resist the reduction in zinc caused by calprotectin. Since GBS alters its own zinc transport mechanisms, upregulating genes encoding zinc-binding proteins, lmb, adcA and adcAII, which can assist GBS to bind zinc.
- Line 158: Instead of "splitting", I would suggest writing "cleaving".
C5a is not a neutrophil chelator, but a potent neutrophil chemoattractant.
Response: Thanks for your good suggestion. "splitting" was changed to "cleaving".
- Line 161: Incomplete sentence: " serotype III accounting for 68.88%" Is it the percentage of invasive or non-invasive isolates or both?
Response: This is a great advice. Now the sentence “Serotype III accounting for 68.88% of all 90 isolates” is complete.
- Line 166: Please define FXIIIA.
Response: Thank you for your good advice. FXIIIA was defined in Line 166.
- Line 178: Correct to: " lack the HylB enzyme."
How does HylB prevent ROS production by neutrophils?
Response: Thank you for your good question. Corrected to: " lack the HylB enzyme." The question was explained with the following point: “Neutrophils produce ROS via Toll-like receptors (TLRs)-2/4 signalling, and pro-inflammatory HA fragments can participate in this signalling, thereby promoting ROS production. Whereas HylB can block ROS production by cleaving HA”.
- Line 243: Explain more about the mechanisms how platelets can kill GBS. And how CPS Sia inhibits synthetic platelet-associated antimicrobial peptide.更改为resist platelet-derived antimicrobial components. In the text: once you write CPS Sia, then GBS Sia. Please be consequent and use the right terminology.
Response: Thanks for your good advice. The mechanism of how platelets can specifically kill GBS is unclear. But how platelets can kill microorganisms was explained with the following points:“Typically, platelets are degranulated by contacting microorganisms through chemotaxis and then using the released kinins and small cationic platelet microbicidal proteins (PMPs) to kill the microorganisms. However, the direct contribution of platelets to the killing of GBS has not been described”. How CPS Sia inhibits synthetic platelet-associated antimicrobial peptide is unclear. Thus, changed related sentence to “Existing studies have shown that CPS Sia can effectively inhibit the killing of GBS by human platelets and resist platelet-derived antimicrobial components. As GBS without Sia expression increased susceptibility to thrombin-activated platelet releasates”. GBS Sia was changed to CPS Sia.
- Line 299 is a repetition of a previous section and should be transferred to the part describing EOD and LOD. Lines 311-320 should also be described in this part.
Response: Thank you for your good advice. I have changed some orders of sections. It is clearer and better now.
- Table 1 is incomplete and should be more concise.
Response: Thank you for your good suggestion. Table 1 was added with prevalence of virulence factors and CAMP factor was deleted as it is not a virulence factor anymore.
- Lines 370-371: ?? 6 Pa??
Response: Thanks for your good discovery. I did not noticed that before. “?? 6 Pa??” was deleted.
Reviewer 2 Report
The authors provide a summary of GBS virulence factors. As a pathogen with significant public health consequences and emerging antimicrobial resistance, this topic is worthy of review. However, this paper has sevearl major flaws.
As virulence factors are discussed, colonization and virulence are frequently confused. This is a key distinction in the development of vaccines and targeted antibiotics. This comes up in two key defects throughout the manuscript: 1)When describing the prevalence of virulence factors in various strains it is unclear if the authors intend to include colonizing isolates or just invasive (and further invasive in neonate? or in ascending infection? or maternal infection?) 2) When describing the mechanism of action of virulence factors it is unclear if the action is in colonization of the vaginal or rectal mucosa, ascendant infection of the upper reproductive tract and fetal tissues, or of the neonate.
The paper reads more as a list of virulence factors and less as a discussion of them. Statements are poorly linked within the paragraphs. Justification and relevance is unclear. Many host and virulence factors are undefined. The effect of virulence factors on some cell types (ie platelets) is well discussed in one section, and then not discussed in other sections where it is highly relevant. Discussion of membrane vesicles is interesting but it is not clear how MVs are behaving differently than live bacteria.
I do want to commend the authors on the tables, they are a very good summary.
Author Response
Response: Thank you very much for your valuable suggestions.
This piece of the review does not focus on colonizing isolates and invasive isolates. Only certain virulence factors that are only expressed in specific strains, such as Srr2, are highlighted when describing them. The article is more specific about whether virulence factors mainly help GBS colonization or invasion.
I have made further changes to address the issue of poorly linked statements within paragraphs and unclear rationale and relevance, which I hope will meet your requirements.
To address the issue that many of the host and virulence factors are not defined, I have refined the terminology as well as the abbreviations that were not clearly explained.
In response to the discussion on platelets, firstly, I would like to thank you for your acknowledgement of my discussion. I have reviewed a lot more of the relevant literature, but the effect of platelets on GBS has not yet been studied clearly. So there is very little literature on how GBS evades platelets. There is only a discussion of platelets in the capsule-related literature, and no relevant articles were found for the other virulence factors. That is why I have only discussed platelets in the section on capsule.
In response to MV's behaviour, first of all, thank you for your acknowledgement of my discussion. Firstly, MV is a vesicle released by GBS and the amount of contents in the vesicle is far less than the virulence factor of GBS. Secondly, the main function of MV is diminishment of the choriodecidual membrane integrity; Infiltration of neutrophils and lymphocytes. whereas GBS does much more than that. Finally, whereas live bacteria have to colonise the surface of the host before they can invade further, MV releases its contents directly to the host for destruction without colonisation itself. Overall, MV is only one of the weapons used by GBS to destroy the host.
Reviewer 3 Report
This manuscript reviews the latest findings of GBS virulence factors related to pathogenesis, with their potential for vaccine development. Generally this manuscript is well written. This reviewer suggests followings to improve this manuscript.
1. Authors listed virulence factors of GBS and explained them. Individual descriptions are OK, however, the prevalence of each virulence factors among pathogenic GBS strains was almost lacking. Although some virulence factors may be present in all the GBS, the others may be distributed to only a portion of GBS (rare virulence factor). Such information of prevalence, i.e., common one or rare ones (or prevalence is unknown) should be added somewhere or descriptions in individual virulence factors.
2. Authors should italicize all the bacterial species names and gene names throughout the manuscript. For example, Streptococcus agalactiae (line 7, 72, 97), pspB (line 117, 118), lmb (line 148). Check whole manuscript.
3. Authors use "virulent factor" (line 43, 57, and many other portions). However, this is strange. All should be corrected as "virulence factor".
4. line 110, the words "Streptococcal Surface Repeat" includes capital letters. It is not sure whether there is any meaning of it. Revise.
5. line 204 should read "ST (sequence type)-17".
6. line 313 should read "CC (clonal complex)-17"
7. In Table 1, column of virulence factor includes either of full names or abbreviated names. This seems to be not well organized and sometimes difficult to collate words in text. For example, in the paragraph of 2.11 GBS vaccine, authors used CPS and ACP. However, in Table 1, they are shown as alpha C protein and capsules, respectively. It is preferable to write full names of virulence factor, and also abbreviated form in parenthesis, such as "Alpha C protein (ACP)".
8. Table 2, column of "Name" may be replaced by "Vaccine candidate" or other.
9. Table 2, column of "Basic factors" may be replaced by "components", "basic components" , or other.
10. line 339, 370, 371: extra stuffs remain. Delete them.
Author Response
- Authors listed virulence factors of GBS and explained them. Individual descriptions are OK, however, the prevalence of each virulence factors among pathogenic GBS strains was almost lacking. Although some virulence factors may be present in all the GBS, the others may be distributed to only a portion of GBS (rare virulence factor). Such information of prevalence, i.e., common one or rare ones (or prevalence is unknown) should be added somewhere or descriptions in individual virulence factors.
Response: Thank you for your good idea about the prevalence of each virulence factors. It is interesting to figure out the prevalence of each virulence factors among 10 serotypes of pathogenic GBS strains. Few paper has reported the prevalence of each virulence factors among 10 serotypes so far. As we know, some virulence factors are common determinants found in all pathogenic serotypes. Some are only expressed in several serotypes, resulting in the dissemination of the pathogenic strain to other tissue such as meningitis in neonates. I have added prevalence in table 2.
- Authors should italicize all the bacterial species names and gene names throughout the manuscript. For example, Streptococcus agalactiae (line 7, 72, 97), pspB (line 117, 118), lmb (line 148). Check whole manuscript.
Response: Thanks for your important advice. All the bacterial species names and gene names throughout the manuscript were italicized.
- Authors use "virulent factor" (line 43, 57, and many other portions). However, this is strange. All should be corrected as "virulence factor".
Response: Thanks for your good reminder. All "virulent factor" were changed to "virulence factor".
- line 110, the words "Streptococcal Surface Repeat" includes capital letters. It is not sure whether there is any meaning of it. Revise.
Response: Thank you for your careful discovery. "Streptococcal Surface Repeat" were changed to “ Streptococcal surface repeat”.
- line 204 should read "ST (sequence type)-17".
Response: Thank you for your good suggestion. I have changed to "ST (sequence type)-17".
- line 313 should read "CC (clonal complex)-17"
Response: Thank you for your good advice. I have changed to"CC (clonal complex)-17".
- In Table 1, column of virulence factor includes either of full names or abbreviated names. This seems to be not well organized and sometimes difficult to collate words in text. For example, in the paragraph of 2.11 GBS vaccine, authors used CPS and ACP. However, in Table 1, they are shown as alpha C protein and capsules, respectively. It is preferable to write full names of virulence factor, and also abbreviated form in parenthesis, such as "Alpha C protein (ACP)".
Response: Thank you for your good comment and advice. All are changed to full names of virulence factor, and also abbreviated form in parenthesis in Table 2 since I added one more table (Table 1) to summarize important guidelines and their general principles for prevention of GBS disease.
- Table 2, column of "Name" may be replaced by "Vaccine candidate" or other.
Response: Thank you for your great advice."Name" were replaced by "Vaccine candidate".
- Table 2, column of "Basic factors" may be replaced by "components", "basic components" , or other.
Response: Thanks for your good suggestion. Column of "Basic factors" were replaced by "basic components".
- line 339, 370, 371: extra stuffs remain. Delete them.
Response: Thanks for your good discovery. Extra stuffs were deleted in line 339, 370, 371.
Round 2
Reviewer 1 Report
The manuscript has been improved. Figure 1 is very nice. English editing required.
Specific comments:
Line 122: There is a mistake" As a pathAs a pAs a pathathogenic bacAs" please correct.
Line 190: Please correct to: "are highly virulent producers of Srr2". And "the CC-1 strain"
Line 191: Correct to "(BJI)", (remove strains after the paranthesis).
Line 199: Instead of "shown", please write "showed"
Line 208: correct to "Srr-1"
Line 400: Add a space before the paranthesis.
Line 523: Please correct to " HylB cleavage of hyaluronic acid" and "enables"
Lines 537-540: The text is not clear. In one place the authors claim that HA fragments promote ROS production, but on the other hand, the authors claim hat the HA fragments produced by HylB-mediated cleavage of HA blocks ROS production. Please explain or correct.
Line 548: "cyl" should be in italics.
Lines 659-660: I think you can remove "from the body". I think you meant: "resists the elimination by mast cells, …etc".
Line 679: Correct to: "protective to the fetus".
Line 804: Please remove "Existing"
Line 807: Spelling mistake " releasates".
Lines 816-818 – It is written in italics, but should be regular letters.
Line 1002 and 1003: I would suggest writing: "is termed EOD" and "is termed LOD".
Line 1034: The bacteria name should be in italics.
Figure 1: The bacterium is depicted with a nucleus, but bacteria do not have a nucleus, so please remove the inner ring.

Author Response
The manuscript has been improved. Figure 1 is very nice. English editing required.
Response: Thanks for your good comment. English editing was finished by a native English-speaking colleague.
Specific comments
Line 122: There is a mistake" As a pathAs a pAs a pathathogenic bacAs" please correct.
Response: Thanks for your great comment. " As a pathAs a pAs a pathathogenic bacAs" was corrected.
Line 190: Please correct to: "are highly virulent producers of Srr2". And "the CC-1 strain"
Response: Thanks for your good advice. Corrected to "are highly virulent producers of Srr2" and "the CC-1 strain".
Line 191: Correct to "(BJI)", (remove strains after the parenthesis).
Response: Thanks for your great advice. “strains” was removed.
Line 199: Instead of "shown", please write "showed"
Response: That is a good advice. Changed to "showed"
Line 208: correct to "Srr-1"
Response: Thanks for your good comment. Corrected to "Srr1"
Line 400: Add a space before the paranthesis.
Response: Sorry I did not find this, though I tried hard. The number of lines you mention is completely different from the number of lines I have seen in the manuscript. The manuscript I saw only had 692 lines in total. I do not know the reason.
Line 523: Please correct to " HylB cleavage of hyaluronic acid" and "enables"
Response: Thanks for your great advice. Corrected to " HylB cleavage of hyaluronic acid" and "enables".
Lines 537-540: The text is not clear. In one place the authors claim that HA fragments promote ROS production, but on the other hand, the authors claim hat the HA fragments produced by HylB-mediated cleavage of HA blocks ROS production. Please explain or correct.
Response: Thanks for your great comment. HA promotes ROS production and disaccharide fragments produced from cleavage of HA inhibit that. I changed related sentence to “Neutrophils produce ROS via Toll-like receptors (TLRs)-2/4 signalling, and pro-inflammatory HA can participate in this signalling, thereby promoting ROS production. Whereas HylB can block ROS production by cleaving HA”.
Line 548: "cyl" should be in italics.
Response: Thanks for your good comment. "cyl" was in italics.
Lines 659-660: I think you can remove "from the body". I think you meant: "resists the elimination by mast cells, …etc".
Response: That is a good advice. "from the body" was removed and changed to "resists the elimination by mast cells, …etc".
Line 679: Correct to: "protective to the fetus".
Response: Thanks for your great advice. Corrected to: "protective to the fetus".
Line 804: Please remove "Existing"
Response: Thanks for your good comment. "Existing" was removed.
Line 807: Spelling mistake " releasates".
Response: Thanks for your discovery. " releasates" was changed to “releases”.
Lines 816-818 – It is written in italics, but should be regular letters.
Response: Sorry I did not find this.
Line 1002 and 1003: I would suggest writing: "is termed EOD" and "is termed LOD".
Response: Thanks for your great advice. Rewritten to "is termed EOD" and "is termed LOD".
Line 1034: The bacteria name should be in italics.
Response: Thanks for your discovery. The bacteria name was in italics.
Figure 1: The bacterium is depicted with a nucleus, but bacteria do not have a nucleus, so please remove the inner ring.
Response: Thanks for your good comment. The nucleus was depicted.
Reviewer 2 Report
The authors addressed reviewer concerns. The additions were valuable.
Author Response
Response: Thanks for your good comments. I will keep on improving my manuscript.